# Mechanistic role of pyroptosis in Kawasaki disease: An integrative bioinformatics analysis of immune dysregulation, machine learning-based biomarker discovery, WGCNA, and drug repurposing insights

Chen Wang[1,2*◉], Qinchao Wu[1,3◉], Jie Chen[2], Jun Wang[1,3], Dan Li[4,5]

**1** Department of Pediatric Internal, Affiliated Hospital of Anhui West Health Vocational College, Lu'an, China, **2** Internal Medicine Teaching and Research Office, Clinical Medicine Department, West Anhui Health Vocational College, Lu'an, China, **3** Nursing Teaching and Research Office, West Anhui Health Vocational College, Lu'an, China, **4** The Lu'an Hospital Affiliated to Anhui Medical University, Lu'an, China, **5** The Lu' an People's Hospital, Lu'an, China

◉ These authors contributed equally to this work.
\* 821014@wahvc.edu.cn

## Abstract

Kawasaki disease (KD) is an acute vasculitis that primarily affects children under five and is a leading cause of acquired heart disease in this age group. Despite the standard treatment with intravenous immunoglobulin (IVIG), approximately 10–20% of patients exhibit IVIG resistance, leading to persistent inflammation and an increased risk of coronary artery aneurysms ( CAA ) . The underlying molecular mechanisms driving KD, particularly the role of pyroptosis, remain incompletely understood. In this study, we employed integrative bioinformatics approaches to investigate the mechanistic role of pyroptosis in KD. By analyzing transcriptomic datasets, we identified differentially expressed genes (DEGs) associated with pyroptosis and immune dysregulation. Weighted Gene Co-Expression Network Analysis (WGCNA) was utilized to uncover key co-expressed gene modules, followed by functional enrichment analyses to explore the biological significance of these genes. Through machine learning-based biomarker discovery, we identified MYD88 and S100A12 as critical pyroptosis-related genes in KD. Their diagnostic potential was validated using external datasets, and their involvement in immune cell infiltration was assessed through computational deconvolution techniques. Furthermore, drug repurposing analysis and molecular docking simulations suggested that Atogepant, Ubrogepant, and Zanubrutinib could serve as potential therapeutic candidates targeting S100A12 and MYD88. These findings provide novel insights into the molecular pathogenesis of KD and highlight potential biomarkers and therapeutic targets for improving KD diagnosis and treatment strategies.

**Data availability statement:** The raw gene expression data supporting the findings of this study are publicly available via GitHub (https://github.com/wangchen-0214/Kawasaki-Pyroptosis-dataexpr.git). A citable version of the dataset has been archived on Zenodo (https://doi.org/10.5281/zenodo.15129021). The code used for data analysis is also available on GitHub (https://github.com/wangchen-0214/Kawasaki-Pyroptosis.git). A citable version of the code has been archived on Zenodo (https://doi.org/10.5281/zenodo.15125948).

**Funding:** The author(s) received no specific funding for this work.

**Competing interests:** The authors have declared that no competing interests exist.

## Introduction

KD is an acute vasculitis that mainly affects children under five. It is a leading cause of acquired heart disease in this age group [1–3]. The exact cause of KD is not clear, but it is believed to result from a mix of genetic factors, environmental triggers, and immune system problems [4–7].

Recent studies suggest that KD may be triggered by viral infections, including Coronavirus Disease 2019 ( COVID-19 ) and its variants. Evidence shows that COVID-19 induces inflammatory responses similar to those seen in KD, such as cytokine release and endothelial injury. COVID-19 has also been linked to Multisystem Inflammatory Syndrome in Children (MIS-C), which shares clinical features with KD. Studies have highlighted the role of cytokines in COVID-19, which may contribute to KD development [8]. Additionally, genetic studies suggest that certain genetic factors in SARS-CoV-2 variants may trigger KD-like symptoms [9].

Treating KD can be difficult because 10–20% of patients do not respond to IVIG, the standard treatment. These IVIG-resistant cases often have ongoing inflammation and a higher chance of developing CAA [3,10,11]. This highlights the importance of understanding the molecular processes involved in KD, which may provide insights into disease progression and treatment strategies.

Bioinformatics provides an effective way to study complex disease mechanisms like those in KD. Recent transcriptomic studies, based on microarray data, demonstrate the potential of analyzing gene expression patterns to uncover important genes and pathways linked to pyroptosis in inflammatory diseases [12,13]. Using bioinformatics tools such as differential gene expression analysis, enrichment analysis, and immune infiltration analysis allows a systematic investigation of pyroptosis in KD. These methods can help identify key pyroptosis-related genes and pathways, paving the way for discovering potential therapeutic targets and biomarkers for diagnosis and treatment.

In recent years, the role of pyroptosis in KD has garnered increasing attention. Studies have shown that pyroptosis, a form of programmed cell death mediated by gasdermins such as Gasdermin D (GSDMD), is accompanied by cell membrane pore formation and the release of inflammatory factors like IL-1β and IL-18 [14–16]. Elevated levels of pyroptosis-related proteins, including ASC, caspase-1, IL-1β, IL-18, and GSDMD, have been observed in the serum of KD patients, suggesting that pyroptosis may play a key role in vascular endothelial injury in KD [16]. In KD mouse models, the use of caspase-1 and NLRP3 inhibitors significantly reduced coronary artery inflammation, further confirming the role of NLRP3-dependent pyroptosis in KD. Additionally, pyroptosis is closely associated with neutrophil infiltration in KD patients [17,18]. Studies have demonstrated that neutrophils activate the NLRP3 inflammasome, triggering caspase-1-dependent pyroptosis and exacerbating vascular inflammation. Although the exact role of pyroptosis in KD pathogenesis remains debated, current research consistently indicates that targeting pyroptosis-related pathways may offer new therapeutic strategies for KD.In conclusion, pyroptosis plays a critical role in the immune response and vascular injury in KD [19,20]. Further bioinformatics analyses and experimental validations are needed to elucidate the specific mechanisms of pyroptosis in KD and to identify potential therapeutic targets.

To bridge these knowledge gaps, this study employs bioinformatics approaches to systematically investigate the link between pyroptosis and KD. Through transcriptomic data analysis, we aim to identify pyroptosis-related genes and pathways that contribute to KD pathogenesis. Further analyses explore immune dysregulation and potential regulatory mechanisms, providing a foundation for identifying novel biomarkers. Additionally, molecular docking and drug prediction analyses were conducted to propose potential therapeutic candidates, advancing efforts to address challenges associated with treatment resistance.

## Methods

### 1. Data acquisition and differential expression analysis

Gene expression matrices were obtained from the GEO database to identify DEGs in KD using whole blood transcriptome datasets. The primary dataset, GSE68004, included 113 samples (37 healthy controls and 76 KD cases), while an independent validation dataset, GSE73461, contained 132 samples (55 healthy controls and 77 KD cases).Additionally, to assess the impact of IVIG treatment on gene expression（GSE16797）, we included data from an independent IVIG treatment cohort. This cohort consisted of paired pre- and post-treatment samples from 6 KD patients, enabling the comparison of gene expression changes before and after IVIG therapy. To ensure consistency and comparability across samples, raw expression data underwent log2 transformation and quantile normalization. Additionally, batch effect correction was applied using the 'sva' package to minimize inter-sample variability. Differential expression analysis was performed using the 'limma' package in R, with DEGs identified based on the criteria of adjusted $P < 0.05$ and $|log2FC| > 0.5$.

### 2. WGCNA

WGCNA was performed to identify gene modules related to KD based on gene expression profiles [21]. After calculating the Pearson correlation coefficient, a similarity matrix was constructed. A soft threshold of 10 was selected to transform the similarity matrix into an adjacency matrix, which was then converted into a topological overlap matrix (TOM). TOM was used for average-linkage hierarchical clustering to classify gene modules, with each module containing at least 50 genes [22]. The "VennDiagram" package was used to visualize the overlap between WGCNA-identified genes and DEGs, and the intersecting genes were defined as Co-Expressed DEGs, representing DEGs within key co-expression modules.

### 3. Identification and functional analysis of co-expressed DEGs

Gene Ontology (GO) enrichment analysis was conducted to explore the biological processes (BP), cellular components (CC), and molecular functions (MF) of DEGs [23]. Kyoto Encyclopedia of Genes and Genomes (KEGG) enrichment analysis was performed to investigate the signaling pathways associated with DEGs [24]. Both GO and KEGG enrichment analyses were conducted using the "clusterProfiler" and "org.Hs.e.g.,db" packages in R.

### 4. Protein-Protein Interaction (PPI) network analysis and identification of hub genes

To further examine the functional relationships among these Co-Expressed DEGs, a PPI network was constructed using the STRING database (https://string-db.org/). Interaction data were retrieved with a confidence score threshold of ≥ 0.7. The network was visualized using Cytoscape software, and key genes within the network were analyzed using the Cyto-Hubba plugin. To identify Hub Genes, the Degree algorithm was applied to assess gene connectivity based on the number of direct interactions. The top-ranked genes with the highest connectivity were selected as Hub Genes, representing potential key regulators in KD.

### 5. Machine learning-based identification of key genes

Three machine-learning algorithms—Least Absolute Shrinkage and Selection Operator (LASSO), Random Forests (RF), and eXtreme Gradient Boosting (XGBoost)—were employed to identify key genes associated with KD. LASSO regression

analysis was conducted using the "glmnet" package, with tenfold cross-validation applied to determine the optimal lambda value. RF analysis was performed using the "randomForest" package, ranking genes based on feature importance scores. XGBoost analysis was conducted using the "XGBoost" package, utilizing gradient boosting to refine key gene selection. The intersection of key genes identified by all three models was considered the final set of machine-learning-derived key genes.

### 6. Evaluation and correlation analysis of infiltration-related immune cells

The Immune Cell Deconvolution Algorithm (CIBERSORT) algorithm was used to analyze immune cell infiltration characteristics in KD and HC groups [25]. CIBERSORT estimates immune cell proportions by applying linear support vector regression to gene expression matrices. The analysis was performed using the "CIBERSORT" package in R, with parameters set as perm = 1000 and QN argument = TRUE.

### 7. Receiver Operating Characteristic (ROC) curve analysis and external validation

ROC curve analysis was performed using the "pROC" package to evaluate the classification performance of the selected key genes. To validate the results, the external dataset GSE73461 was used as an independent cohort. Furthermore, transcriptomic data from six paired KD patients (GSE16797) were analyzed to examine changes in MYD88 and S100A12 expression before and after IVIG treatment. Differential expression analysis was conducted using the "limma" package, and the results were visualized with boxplots.

### 8. Identification of key-genes and gene set enrichment analysis (GSEA)

To investigate the enrichment of Key-genes in hallmark gene sets, GSEA was performed using the "clusterProfiler" package in R [26]. The analysis focused on 50 hallmark pathways from the Molecular Signatures Database (MSigDB). Differential expression metrics were used to rank the gene list, and pathways with adjusted p-values (P.adjust < 0.05) were considered significant. Four significant pathways were identified, and enrichment plots were used to visualize the Running Enrichment Score (RES) and the distribution of Key-genes across ranked gene lists.

### 9. Comprehensive analysis of key-genes: drug prediction and molecular docking

Drugs targeting the Key-genes were identified using the Drug-Gene Interaction Database (DGIdb) (https://dgidb.org) to explore potential therapeutic agents [27]. The crystal structures of Key-genes were obtained from the Uniprot (https://www.uniprot.org), while the 3D structures of small molecule drugs were downloaded from the PubChem database (https://pubchem.ncbi.nlm.nih.gov/). The drug structures were handle using PyMOL software, and molecular docking was performed using CB-Dock (https://cadd.labshare.cn/cb-dock2/php/blinddock.php) [28].

## Results

### 1. Screening of co-expressed DEGs in KD

**Identification of DEGs.** Using the preprocessed gene expression matrix from the GSE68004 dataset, a total of 514 DEGs were identified between patients with KD and healthy controls. DEGs were screened based on the criteria of adjusted P-value < 0.05 and |log2FC| > 1. A volcano plot (Fig 1A) was generated using the R package ggplot2 to visualize these DEGs, clearly highlighting upregulated and downregulated genes.

**WGCNA.** To explore gene co-expression patterns associated with pyroptosis in KD, WGCNA was performed to construct a gene co-expression network and identify modules significantly correlated with KD. The R package WGCNA was employed for this analysis. First, hierarchical clustering based on the average linkage method and Pearson

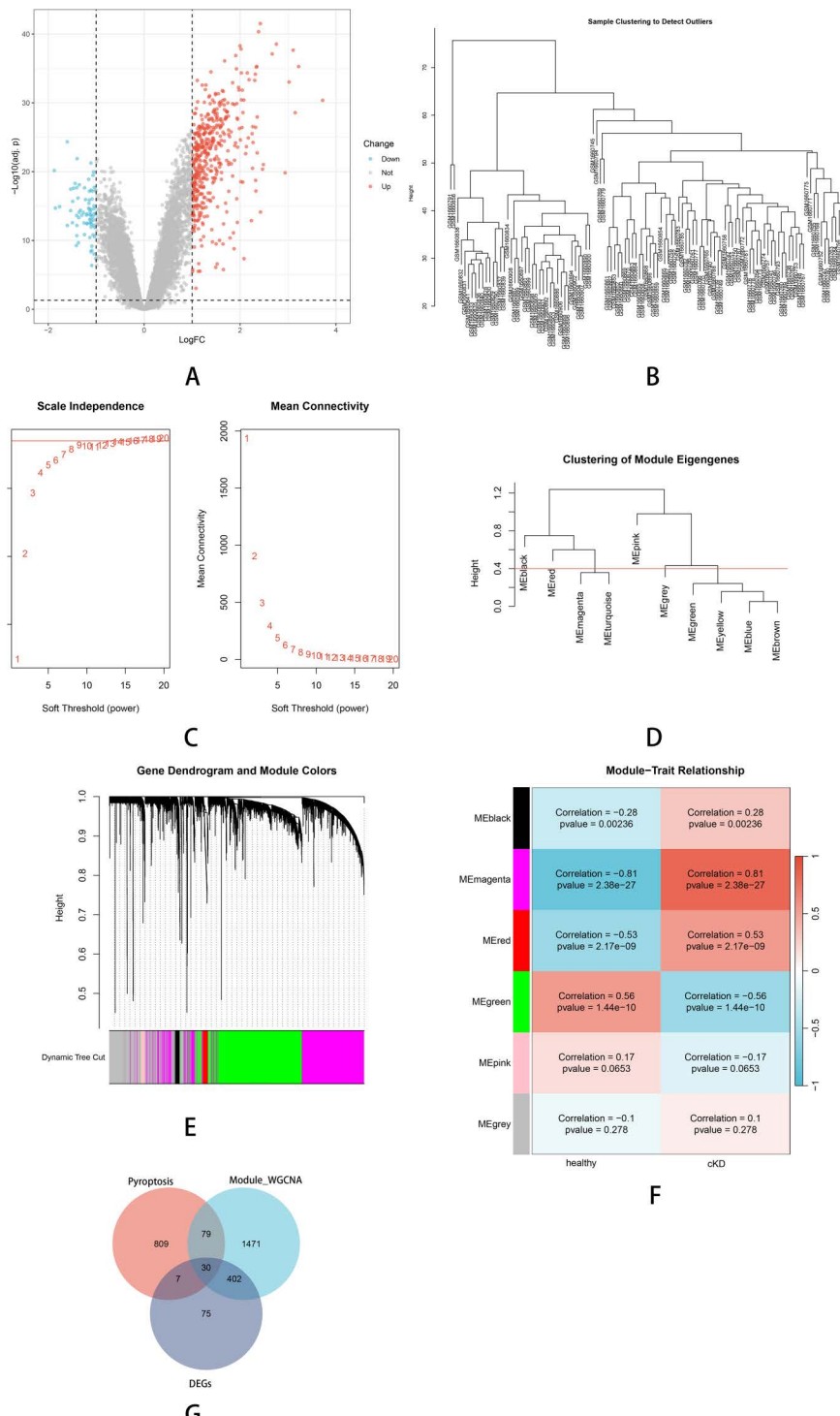

**Fig 1. Screening of Co-Expressed DEGs in KD.** (A) Volcano Plot of DEGs in KD(GSE68004). (B) Sample Clustering and Outlier Detection. (C) Soft Threshold Power Selection for WGCNA. (D) Clustering of Module Eigengenes. (E) Gene Dendrogram and Module Identification. (F) Module-Trait Relationships in KD. (G) Venn diagram highlighting overlaps of DEGs(purple), pyroptosis-related genes (red, retrieved from external databases), and WGCNA genes(green).

correlation was conducted to detect outlier samples, which were subsequently removed to ensure the reliability of the data (Fig 1B). A soft thresholding power (β) of 10 was selected to achieve scale-free topology, as determined by a scale independence value greater than 0.75 (Fig 1C). Initially, hierarchical clustering analysis and dynamic TreeCut methods identified ten distinct gene modules (Fig 1D and 1E). To enhance the robustness and interpretability of the network, similar modules were merged based on eigengene similarity, yielding six final modules for further analysis. Among these, the green (correlation = 0.81, P-value = 2.38e-27) and pink modules (correlation = 0.56, P-value = 1.44e-10) were identified as key modules, exhibiting significant correlations with KD (Fig 1F).

**Identification of co-expressed DEGs.** A total of 925 genes associated with the regulation of pyroptosis were retrieved from the GeneCards database (https://www.genecards.org/). These pyroptosis-related genes, curated from diverse data sources, encompass those involved in pyroptosis regulation, signaling, and associated metabolic pathways. To explore key genes potentially implicated in the pathogenesis of KD through pyroptosis, an overlap analysis was conducted among the previously identified DEGs, genes from WGCNA-derived core modules, and pyroptosis-related genes. The intersection analysis identified 30 Co-Expressed DEGs common to these gene sets (Fig 1G).

## 2. Enrichment analysis of co-expressed DEGs

GO enrichment analysis indicated that the identified hub genes were significantly enriched in several biological processes (BP), including neutrophil activation, positive regulation of cytokine production, response to lipopolysaccharide, and immune response-regulating signaling pathway. Additionally, enriched cellular components (CC) included specific granule lumen and primary lysosome, and enriched molecular functions (MF) included immune receptor activity, cytokine receptor activity, and lipopolysaccharide binding. These findings were visualized using a dot plot (Fig 2A) and a circular plot displaying log fold change (Fig 2B), highlighting the key GO terms enriched among the Co-Expressed DEGs.

KEGG pathway enrichment analysis further revealed that these Co-Expressed DEGs were predominantly involved in pathways related to immune regulation and inflammation, such as the NF-kappa B signaling pathway, Legionellosis, and pathways associated with inflammatory responses. The top significantly enriched pathways were visualized using a bar chart (Fig 2C), providing clear insights into their functional relevance in KD and pyroptosis.

Collectively, these results suggest that the identified hub genes potentially regulate pyroptosis-associated immune responses and inflammatory pathways, playing critical roles in the pathogenesis and progression of KD.

## 3. Identification and expression validation of hub genes in KD

To explore key regulatory elements involved in pyroptosis associated with KD, a PPI network analysis was performed. The STRING database was utilized to construct the network, and Cytoscape software was employed for visualization. Nodes within the network were ranked based on their degree of connectivity, reflecting their importance. The PPI network of Co-Expressed DEGs is shown in Fig 3A, with hub genes highlighted in warmer colors. The top ten highly connected genes, identified as Hub Genes, were IL1B, TLR4, MYD88, MMP9, S100A12, CAMP, LY96, LCN2, FPR2, and MAPK14. The top 10 Hub Genes subnetwork, ranked by degree of connectivity, is illustrated in Fig 3B. These Hub Genes exhibited robust interactions, suggesting their central roles in mediating pyroptosis-related processes in KD.

The differential expression of these top-ranked genes between KD patients and healthy controls was evaluated. Boxplots in Fig 3C clearly demonstrated that these genes exhibited significantly altered expression levels in KD compared with healthy individuals.

## 4. Identification of key genes

To identify key genes involved in pyroptosis regulation associated with KD, three machine learning algorithms—LASSO, RF, and XGBoost—were employed. The LASSO regression analysis, performed using the R package glmnet, identified

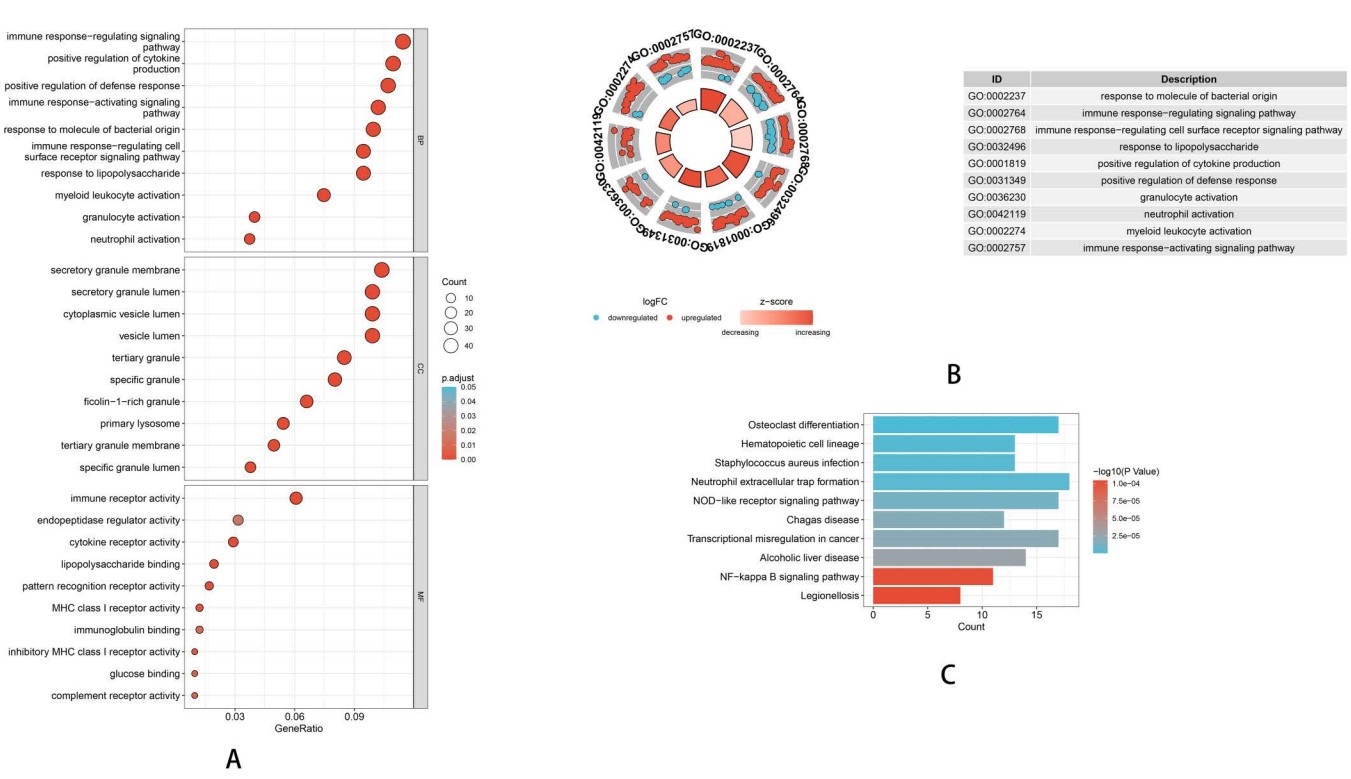

**Fig 2. Enrichment Analysis of Co-Expressed DEGs.** (A) GO Enrichment Analysis of Co-Expressed DEGs (Dot Plot). (B) Log Fold Change (logFC) Visualization of GO Enrichment. (C) KEGG Pathway Enrichment Analysis of Co-Expressed DEGs (Bar Plot).

five key genes after cross-validation determined the optimal parameter λ (Fig 4A). The coefficient trajectory of the selected genes across different λ values is shown (Fig 4B).The RF algorithm, executed with the randomForest package, ranked genes according to their importance scores based on Mean Decrease Gini, selecting the top five genes (Fig 4C).The feature importance ranking of genes based on the RF model is illustrated (Fig 4D).Similarly, XGBoost, implemented through the XGBoost package, identified five important genes based on feature importance scores (Fig 4E).The SHAP summary plot highlights the contribution of each gene to the XGBoost model (Fig 4F).The intersection analysis across these three machine learning methods ultimately highlighted two central key genes, MYD88 and S100A12, shared among these analyses (Fig 4G).

## 5. Expression analysis and correlation of key genes

To validate the diagnostic performance of the identified key pyroptosis-related genes (MYD88 and S100A12) in KD, ROC curve analyses were conducted. The analysis demonstrated excellent diagnostic accuracy in distinguishing KD patients from healthy controls. Specifically, the area under the ROC curve (AUC) values for MYD88 and S100A12 were 0.994 and 0.999, respectively, in the training dataset (Fig 5A and 5B), indicating outstanding sensitivity and specificity.

Furthermore, external validation was performed using the independent dataset (GSE73461), confirming the robustness of these findings. The AUC values for MYD88 and S100A12 were 0.926 and 0.961, respectively, further highlighting their strong potential as diagnostic biomarkers for KD-associated pyroptosis (Fig 5C and 5D). These results strongly support the potential utility of MYD88 and S100A12 as biomarkers or therapeutic targets in KD, warranting further functional and clinical validation studies.

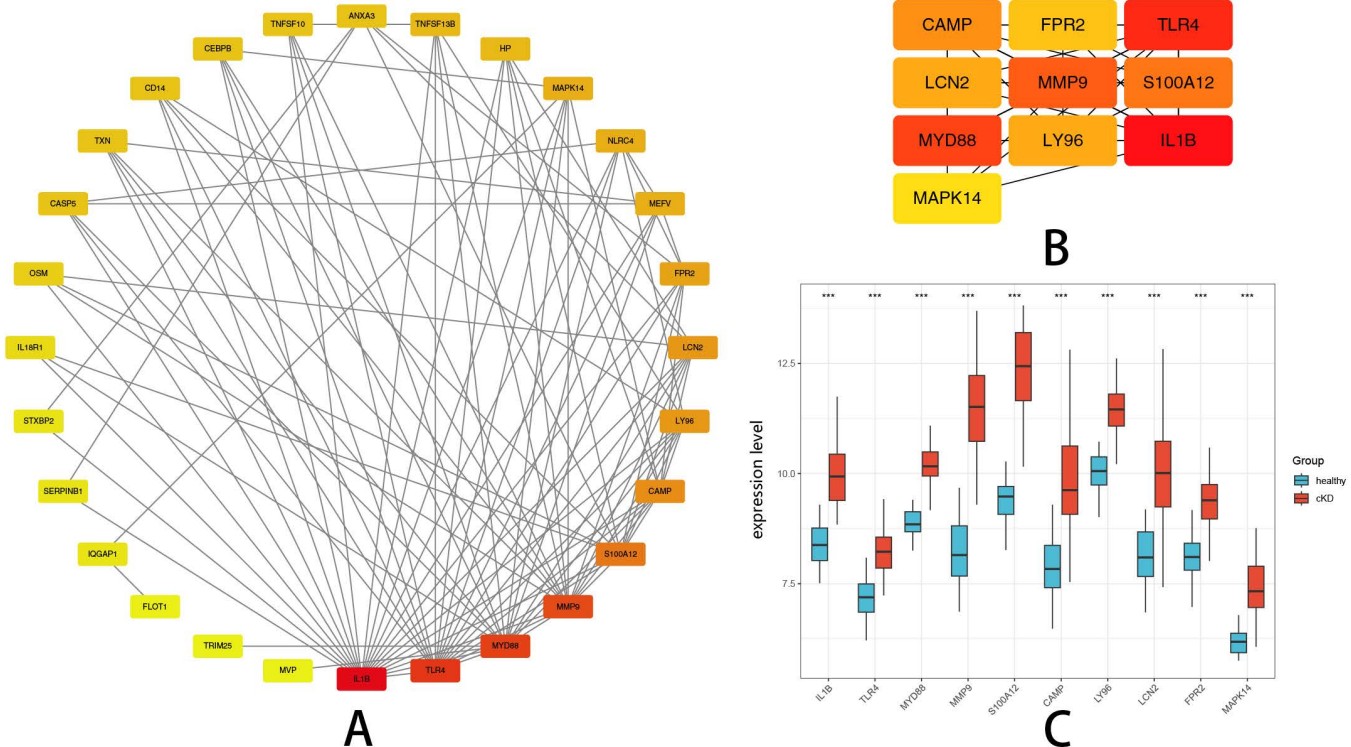

**Fig 3. Identification and Expression Validation of Hub Genes in KD.** (A) PPI network visualization of co-expressed DEGs. (B) Top 10 hub genes identified from PPI network. (C) Boxplots illustrating expression levels of hub genes in KD compared to controls.

To further explore the clinical relevance of these genes, we analyzed their expression changes before and after IVIG treatment in IVIG-responsive KD patients using dataset GSE16797. Notably, both MYD88 and S100A12 showed significantly decreased expression following IVIG therapy, suggesting their involvement in the acute inflammatory phase and response to treatment (Fig 5E and 5F). These findings provide additional transcriptomic evidence supporting the potential of MYD88 and S100A12 as biomarkers not only for diagnosis but also for treatment monitoring in KD.

## 6. GSEA reveals functional pathways of key genes

GSEA was performed to evaluate the enrichment of hallmark gene sets from the MSigDB associated with the two identified key genes (MYD88 and S100A12). The analysis considered 50 hallmark pathways, with several pathways significantly enriched for each gene.

For MYD88, significantly enriched pathways included TNFα signaling via NF-κB, Inflammatory Response, Apoptosis, and Hypoxia(Fig 6A and 6B). These results underline the pivotal role of MYD88 in immune-driven inflammation, which are key elements in KD-associated pyroptosis.

For S100A12, significantly enriched pathways included IL2-STAT5 signaling, Oxidative phosphorylation, Inflammatory response, and TNFα signaling via NF-κB(Fig 6C and 6D). Particularly, the enrichment of pathways such as inflammatory response and TNFα signaling emphasizes its potential involvement in immune regulation and inflammatory processes in KD.

These results highlight the functional significance of MYD88 and S100A12 in regulating inflammation and immune responses critical to KD pathogenesis through pyroptotic mechanisms, further supporting their potential as biomarkers or therapeutic targets.

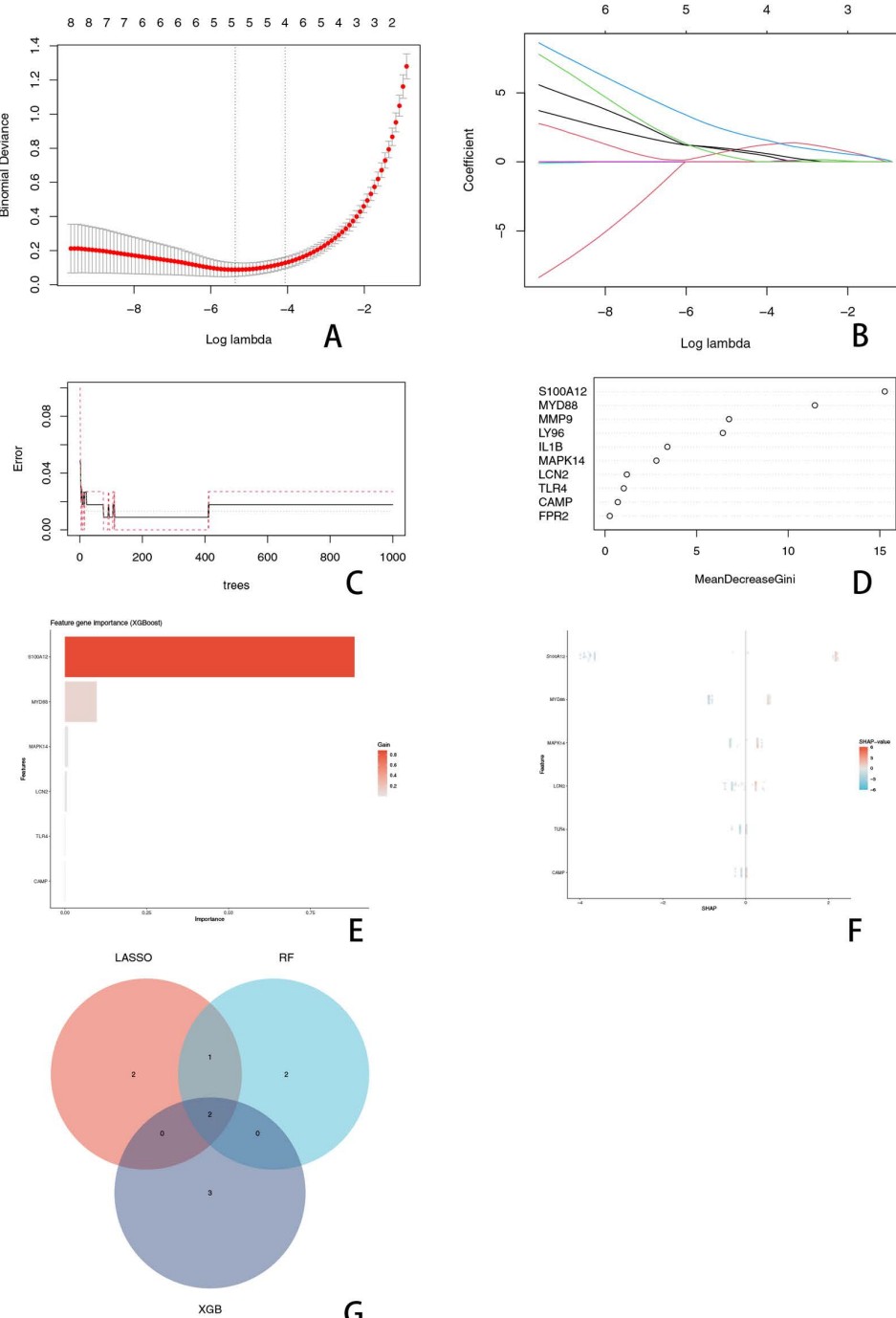

**Fig 4. Identification of Key Genes.** (A) illustrates the binomial deviance plot, where the optimal λ value was selected based on cross-validation to refine the key gene set.(B) Shows the path of gene selection in LASSO regression, highlighting the coefficients of the selected genes along the regularization path. (C) Error rate curve of the RF model as the number of decision trees increases, indicating model stability. (D) Feature importance ranking in the RF model based on Mean Decrease Gini, highlighting the most influential genes. (E) Feature importance analysis in the XGBoost model, where gene importance is measured using Gain values. (F) SHAP value summary plot for the XGBoost model, illustrating the contribution and directional impact of each gene on model predictions. (G) Venn diagram showing the intersection of key genes identified by LASSO, RF, and XGBoost, emphasizing MYD88 and S100A12 as the common critical genes.

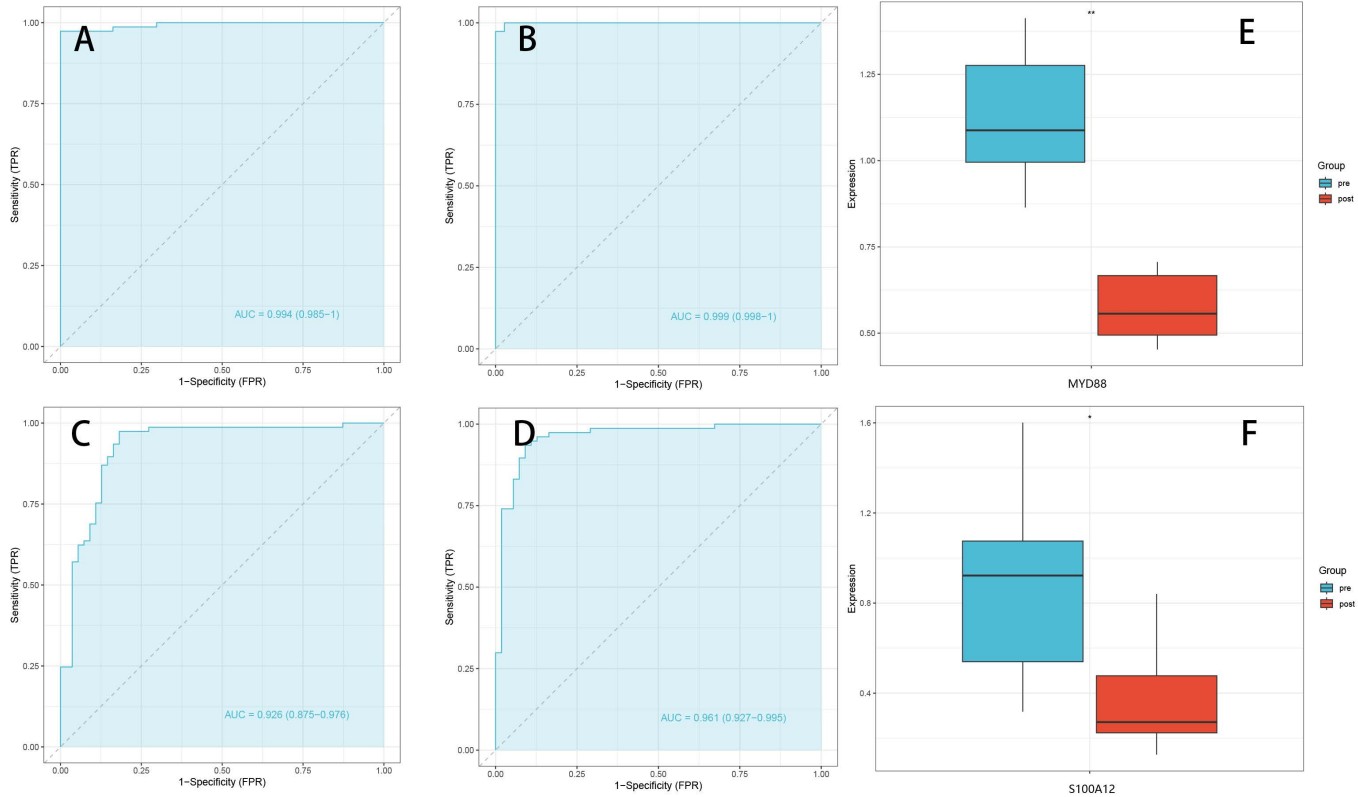

**Fig 5. Expression Analysis and Correlation of Key Genes.** (A)ROC curve of MYD88 in the training dataset, showing an AUC of 0.994, indicating high diagnostic accuracy. (B) ROC curve of S100A12 in the training dataset, with an AUC of 0.999, reflecting nearly perfect discrimination between KD patients and healthy controls. (C) External validation ROC curve for MYD88 in the GSE68004 dataset, with an AUC of 0.926, supporting its reliability as a biomarker. (D) External validation ROC curve for S100A12 in the GSE68004 dataset, showing an AUC of 0.961, further confirming its diagnostic potential. (E) Expression levels of MYD88 in IVIG-responsive Kawasaki disease patients (pre vs. post IVIG) ( GSE16797 ) . (F) Expression levels of S100A12 in IVIG-responsive Kawasaki disease patients (pre vs. post IVIG) ( GSE16797 ) .

## 7. Immune infiltration and key gene correlations

To explore the immune cell infiltration patterns associated with KD and pyroptosis, immune infiltration analyses were performed using the CIBERSORT algorithm. Immune cell composition was visualized using stacked bar plots, and differential infiltration levels between KD patients and healthy controls were compared using boxplots.

The stacked bar plot illustrated the overall immune cell composition between groups (Fig 7A), while the boxplot indicated significantly increased infiltration of immune cell populations, including activated neutrophils, monocytes, activated CD4 memory T cells, and activated NK cells in KD compared to healthy controls, suggesting their critical roles in disease progression and pyroptosis-related inflammation (Fig 7B).

To further evaluate the relationships between the identified key genes (MYD88 and S100A12) and immune cell populations, correlation analyses were conducted. Lollipop plots illustrated significant correlations between gene expression levels and immune infiltration.

MYD88 exhibited significant positive correlations with Neutrophils. MYD88 showed significant negative correlations with T cells CD4 naive, NK cells activated, T cells CD8, T cells regulatory (Tregs) and macrophages M1 (Fig 7C). S100A12 exhibited significant negative correlations with M1 macrophages, Tregs,NK cells activated,T cells CD4 naive and T cells CD8 (Fig 7D).

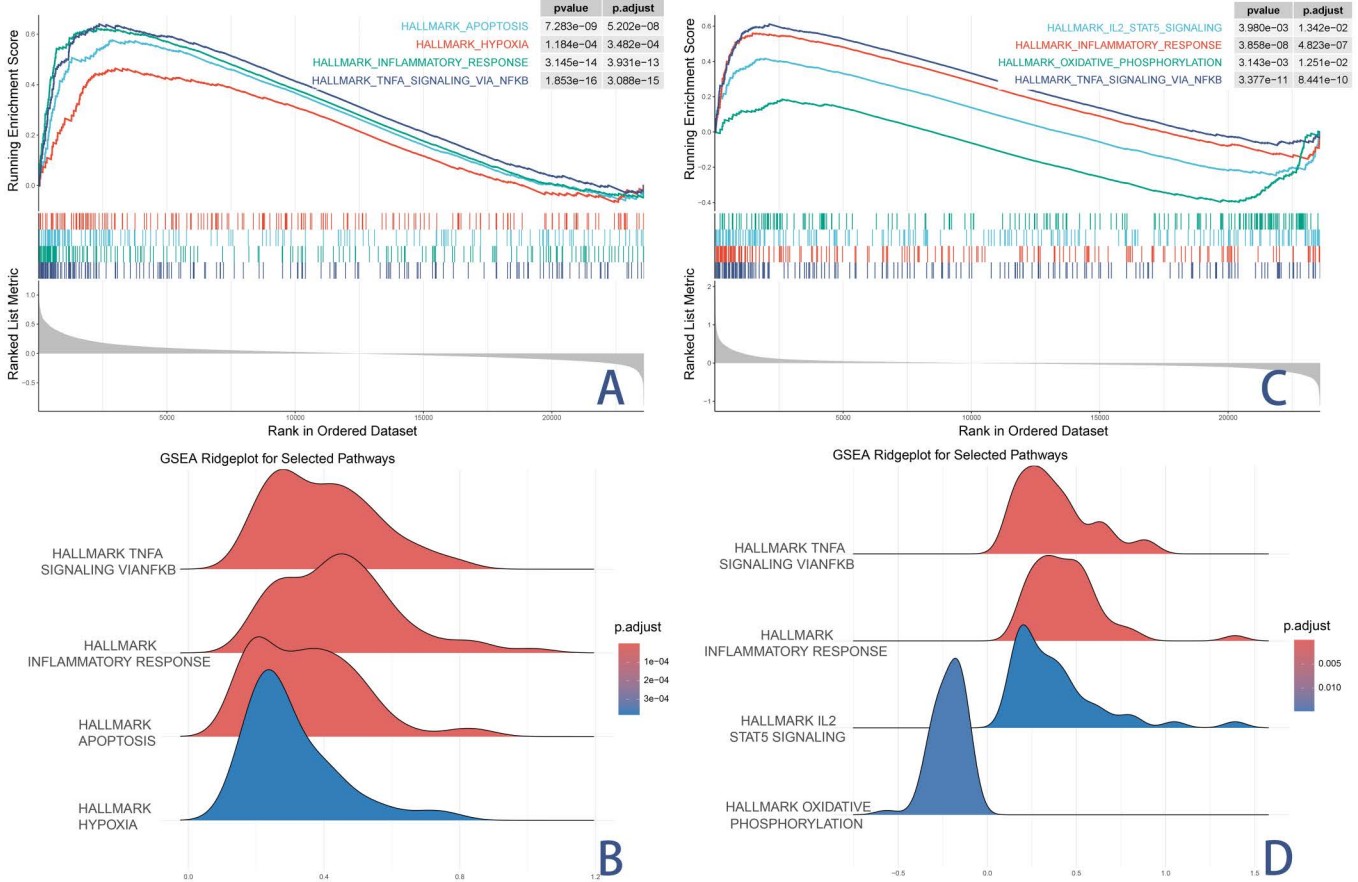

**Fig 6. GSEA Reveals Functional Pathways of Key Genes.** (A) shows the GSEA enrichment plot for MYD88, demonstrating its association with inflammatory and apoptotic pathways. (B) Ridge plot for the top hallmark pathways enriched in MYD88, showing their respective adjusted p-values. (C) presents the GSEA results for S100A12, confirming its involvement in immune regulation and oxidative stress. (D) Ridge plot for the top hallmark pathways enriched in S100A12, displaying the adjusted p-values for each pathway.

These findings highlight the potential roles of MYD88 and S100A12 in modulating immune responses and inflammatory microenvironments through pyroptosis pathways in KD, providing insights into their potential clinical significance as biomarkers or therapeutic targets.

## 8. Drug prediction and molecular docking

To identify potential therapeutic agents targeting the key regulators (S100A12 and MYD88) involved in KD-associated pyroptosis, a drug screening followed by molecular docking analyses was conducted.

For S100A12, two candidate small-molecule drugs, Atogepant and Ubrogepant, were identified. Docking analyses demonstrated that Atogepant exhibited strong binding affinity (−8.5 kcal/mol) with critical amino acid residues (LEU35, GLU39, LEU40, and ILE13), indicating stable interactions and potential biological activity (Fig 8A and 8B).

Similarly, Ubrogepant showed robust interactions (−8.1 kcal/mol) with residues including LEU35, LYS38, GLU39, LEU40, and PHE73 (Fig 8C and 8D), suggesting its possible regulatory effects on the S100A12 protein.

For MYD88, molecular docking identified Zanubrutinib as a promising therapeutic candidate with a significant binding affinity (−7.9 kcal/mol). Docking results highlighted substantial interactions with key residues such as PHE164, ILE165,

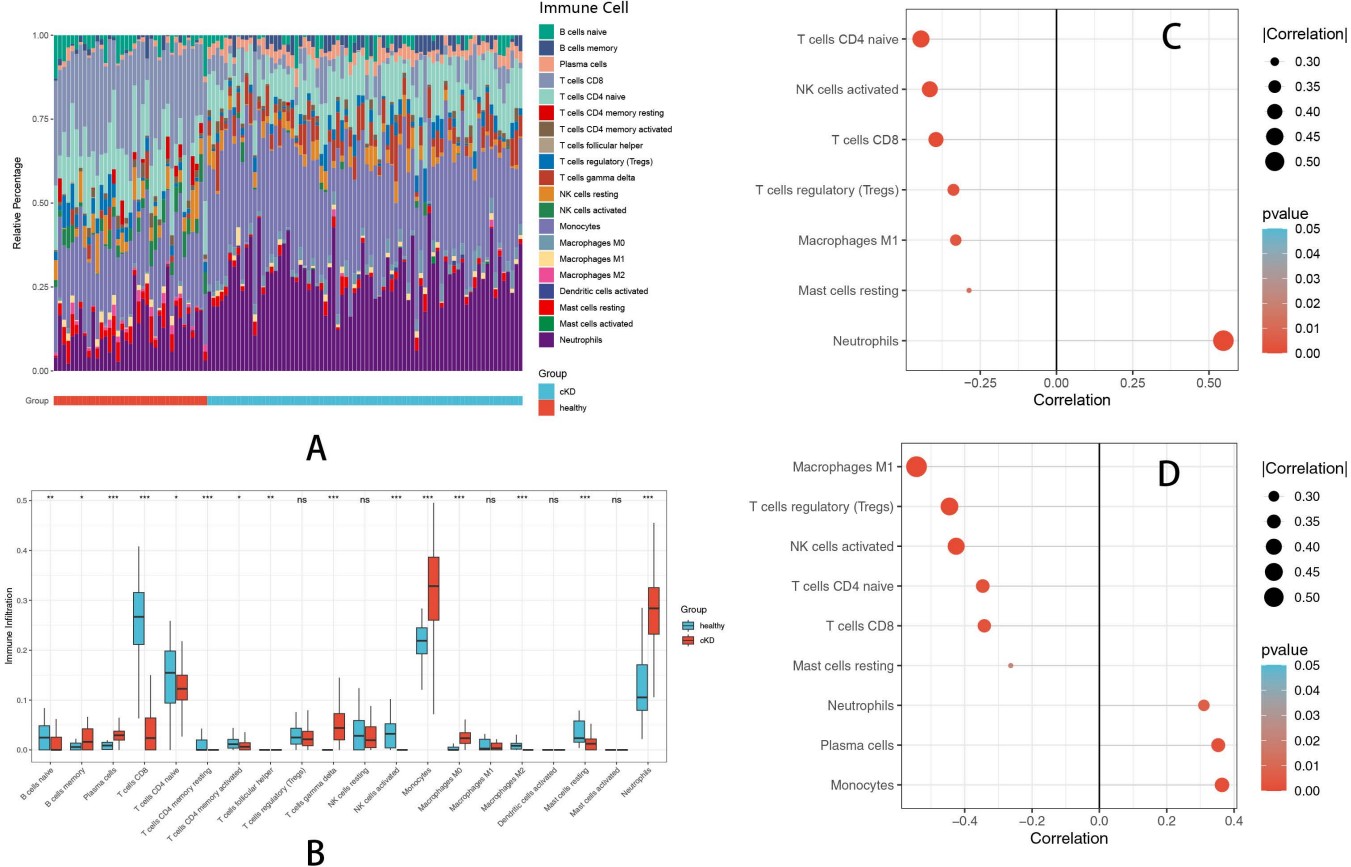

**Fig 7. Immune Infiltration and Key Gene Correlations.** (A)depicts the distribution of immune cell subsets, showing increased immune activation in KD. (B) visualizes differential immune infiltration, indicating that these immune cell subsets may contribute to pyroptosis-driven inflammation in KD. (C) displays a lollipop plot illustrating the correlation between MYD88 expression and immune infiltration, emphasizing its pro-inflammatory role in KD. (D) highlights the immune cell correlations of S100A12, linking it to T cell regulation and inflammatory pathways in KD.

CYS166, TYR167, CYS168, PRO169, ILE172, SER194, ASP195, VAL198, and SER209 within the active site (Fig 8E and 8F), supporting its potential to modulate MYD88-mediated inflammatory and pyroptotic pathways.

Collectively, these findings suggest that Atogepant, Ubrogepant, and Zanubrutinib could serve as potential therapeutic candidates for KD through their targeted modulation of S100A12- and MYD88-associated pyroptotic processes. Further experimental validation and clinical evaluations are warranted to confirm their therapeutic efficacy and practical applications.

## Discussion

This study highlights MYD88 and S100A12 as crucial regulators in KD-associated pyroptosis. Both genes showed significant differential expression and were central within the constructed protein–protein interaction network, indicating their potential as pivotal biomarkers. Functional enrichment analyses linked these genes to inflammation-related pathways, notably TNFα signaling via NF-κB, IL2-STAT5 signaling, and the general inflammatory response. These pathways underscore the integral roles of innate and adaptive immune interactions driving inflammation and vascular damage in KD, aligning with known disease mechanisms [29,30].MYD88, an essential adaptor protein in Toll-like receptor (TLR) and interleukin-1 receptor signaling, plays an important role in recognizing pathogen-associated molecular patterns, such as

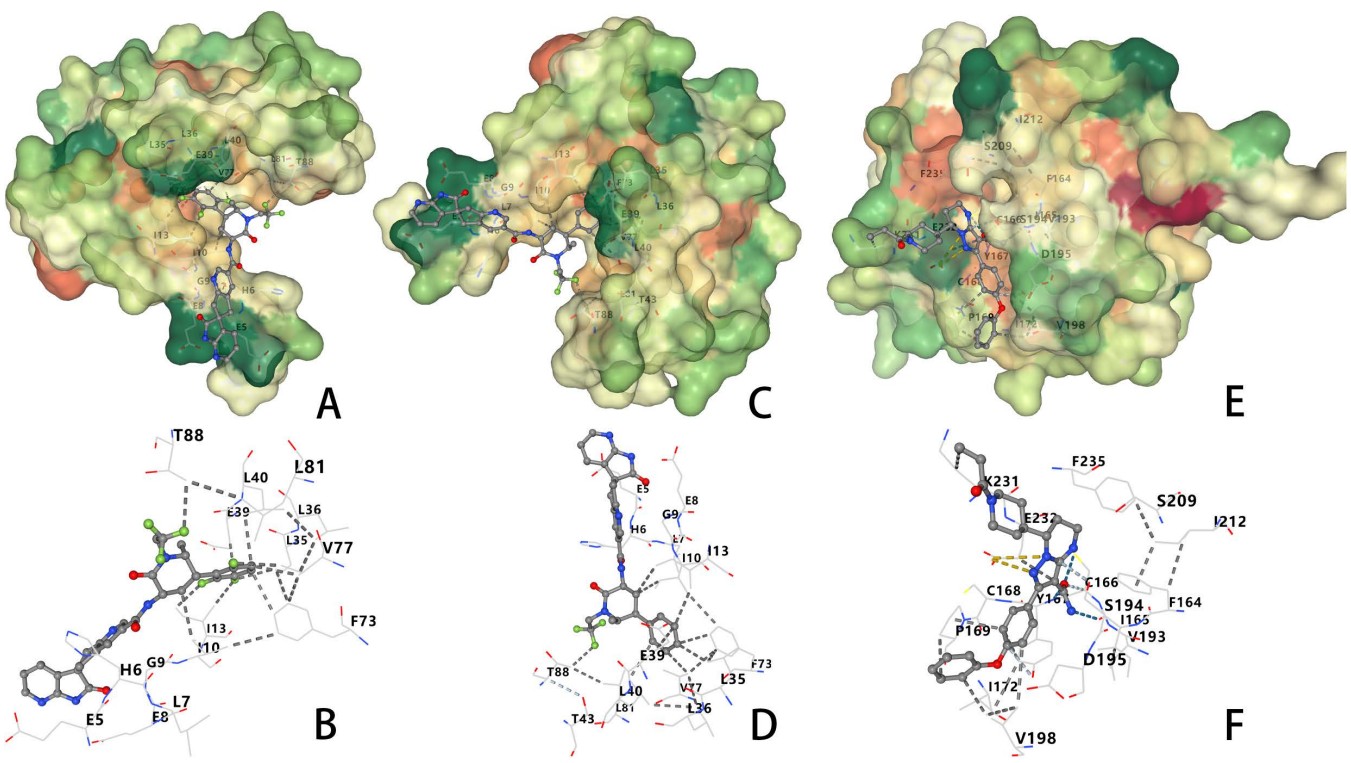

**Fig 8. Drug Prediction and Molecular Docking.** (A) Molecular docking results of Atogepant with S100A12, showing strong binding interactions. (B) Detailed view of Atogepant's binding interactions with S100A12, highlighting key amino acid residues involved. (C) Molecular docking results of Ubrogepant with S100A12, displaying its potential as a therapeutic candidate. (D) Close-up view of Ubrogepant's binding with S100A12, showing its key interacting residues. (E) Molecular docking results of Zanubrutinib with MYD88, illustrating significant binding affinity. (F) Detailed docking visualization of Zanubrutinib's interaction with MYD88, highlighting key amino acid residues.

viral nucleic acids, and triggering intracellular signaling pathways that culminate in the activation of interferon regulatory factor 3 (IRF3) and NF-κB [31]. Previous studies have reported that MYD88 is significantly upregulated in the acute phase of KD, contributing to excessive pro-inflammatory cytokine production via NF-κB activation [32,33]. Likewise, S100A12, a member of the S100/calgranulin family secreted by activated neutrophils, has been identified as a potential biomarker for KD disease activity and coronary artery lesion risk [34,35]. The upregulation of MYD88 and S100A12 observed in our study is consistent with these findings, providing additional transcriptomic evidence supporting their relevance in KD pathogenesis.

Immune infiltration analysis further revealed significant correlations between MYD88 and S100A12 expression and various immune cell populations, such as macrophages, regulatory T cells (Tregs), and activated NK cells. This suggests these genes may actively shape the immune microenvironment in KD, potentially mediating pyroptotic inflammation. Molecular docking analyses identified promising therapeutic candidates—Atogepant, Ubrogepant, and particularly Zanubrutinib—demonstrating strong binding affinities to S100A12 and MYD88-related proteins, thus offering potential therapeutic interventions through repurposing established drugs. Notably, Zanubrutinib, a second-generation Bruton's Tyrosine Kinase (BTK) inhibitor, has shown potential in modulating MYD88-mediated inflammatory pathways. Since BTK plays a critical role in B-cell receptor (BCR) signaling and innate immune regulation, inhibiting BTK may provide a dual benefit by suppressing aberrant immune activation and dampening pyroptosis-driven inflammation in KD. Given that MYD88-dependent pathways intersect with BTK signaling, targeting BTK alongside MYD88 could represent a promising strategy for mitigating excessive immune responses and vascular damage in KD pathogenesis [36–39].

Moreover, the application of integrative bioinformatics and machine learning methodologies in our study represents a novel approach in KD research, allowing comprehensive identification of biomarkers and potential drug candidates. Unlike earlier works that focused on single cytokines or isolated pathways, our systems-level analysis provides a broader understanding of the complex molecular interactions driving pyroptosis in KD and offers tangible targets for clinical translation.

While our results offer valuable insights, this study has several inherent limitations.It relied primarily on publicly available datasets, which may have inherent variability due to limited sample sizes and patient heterogeneity. Moreover, detailed clinical characteristics of the patients were not fully available, limiting the ability to assess how these clinical factors might contribute to differences in gene expression profiles. Additionally, the computational predictions of biomarker importance and drug-target interactions require experimental validation to confirm their biological relevance and therapeutic efficacy.Importantly, our analysis was conducted only at the transcriptomic level, and we did not perform protein-level validation such as Western blotting or immunohistochemistry. Future studies incorporating protein expression analyses would greatly enhance the robustness of our findings. Another limitation is the correlative nature of our immune infiltration findings, which cannot definitively establish causality between key genes and immune responses. Furthermore, molecular docking results, while promising, necessitate verification through biological experiments to assess true clinical potential.

Future studies should focus on experimental validation of MYD88 and S100A12 functions, utilizing in vitro and in vivo models to confirm their roles in pyroptosis and inflammation in KD. Investigating these genes' mechanistic involvement in modulating immune cell interactions could further elucidate disease pathways and identify additional therapeutic targets. Additionally, preclinical studies assessing the therapeutic efficacy of Atogepant, Ubrogepant, and Zanubrutinib in KD models are crucial next steps. Given their established safety profiles for other indications, positive findings could rapidly advance these drugs into clinical trials, providing urgently needed therapeutic alternatives for KD patients resistant to standard treatments.

## Conclusion

This integrative bioinformatics analysis highlights the critical roles of MYD88 and S100A12 in KD-associated pyroptosis, underscoring their potential as diagnostic biomarkers and therapeutic targets. Functional enrichment analyses revealed significant involvement of inflammatory and immune-related pathways, particularly TNFα signaling via NF-κB, IL2-STAT5 signaling, oxidative phosphorylation, and apoptosis, emphasizing their mechanistic relevance in disease pathogenesis. Additionally, molecular docking analyses identified promising therapeutic candidates (Atogepant, Ubrogepant, and Zanubrutinib), warranting further validation studies. These findings support the development of novel therapeutic interventions for KD, warranting further preclinical and clinical investigations to accelerate drug repurposing strategies.

## Author contributions

**Conceptualization:** Chen Wang, Qinchao Wu.

**Data curation:** Jie Chen.

**Formal analysis:** Jie Chen, Jun Wang.

**Methodology:** Qinchao Wu.

**Project administration:** chen wang.

**Visualization:** Jun Wang, Dan Li.

**Writing – original draft:** Chen Wang.

**Writing – review & editing:** Chen Wang, Qinchao Wu, Jie Chen, Jun Wang, Dan Li.

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
