## [Decision Letter · Decision Letter 0]

31 Mar 2025

PONE-D-25-12866Mechanistic Role of Pyroptosis in Kawasaki disease: An Integrative Bioinformatics Analysis of Immune Dysregulation, Machine Learning-Based Biomarker Discovery, WGCNA, and Drug Repurposing InsightsPLOS ONE

Dear Dr. wang,

Thank you for submitting your manuscript to PLOS ONE. After careful consideration, we feel that it has merit but does not fully meet PLOS ONE’s publication criteria as it currently stands. Therefore, we invite you to submit a revised version of the manuscript that addresses the points raised during the review process.

**ACADEMIC EDITOR's COMMENTS:** 

1. The authors failed to introduce that COVID and variants of concerns can cause KD. The authors should introduce this. More references should be cited, with these two as an example (citing is optional):

Liu BM, Martins TB, Peterson LK, Hill HR. Clinical significance of measuring serum cytokine levels as inflammatory biomarkers in adult and pediatric COVID-19 cases: A review. Cytokine. 2021 Jun;142:155478. doi: 10.1016/j.cyto.2021.155478. Epub 2021 Feb 23. PMID: 33667962; PMCID: PMC7901304.

Liu BM, Yao Q, Cruz-Cosme R, Yarbrough C, Draper K, Suslovic W, Muhammad I, Contes KM, Hillyard DR, Teng S, Tang Q. Genetic Conservation and Diversity of SARS-CoV-2 Envelope Gene Across Variants of Concern. J Med Virol. 2025 Jan;97(1):e70136. doi: 10.1002/jmv.70136. PMID: 39744807.

2. The authors should introduce the function of Myd88. Pattern recognition receptors, such as Myd88, play important roles in recognizing viral nucleic acids as a major class of viral pathogen-associated molecular pattern and triggering intra cellular signaling pathways that culminate in activation of interferon regulatory factor 3 (IRF3) and NF-κB. More references should be cited, with this one (PMID: 38556084) as an example (citing is optional)

We look forward to receiving your revised manuscript.

Kind regards,

Benjamin M. Liu, MBBS, PhD, D(ABMM), MB(ASCP)

Academic Editor

PLOS ONE

Reviewers' comments:

Reviewer's Responses to Questions

**Comments to the Author**

1. Is the manuscript technically sound, and do the data support the conclusions?

Reviewer #1: Yes

Reviewer #2: Yes

2. Has the statistical analysis been performed appropriately and rigorously? 

Reviewer #1: Yes

Reviewer #2: Yes

3. Have the authors made all data underlying the findings in their manuscript fully available?

Reviewer #1: No

Reviewer #2: Yes

4. Is the manuscript presented in an intelligible fashion and written in standard English?

Reviewer #1: Yes

Reviewer #2: Yes

5. Review Comments to the Author

Reviewer #1: With the maturity of bioinformatics research ideas and technical routes, network pharmacology analysis can be performed prior to cell and animal experiments to screen potential drug candidates and provide potential therapeutic targets for disease-related molecular mechanisms as subsequent empirical studies.

The paper first constructs a gene co-expression network by identifying differentially expressed genes between Kawasaki disease patients and healthy populations, and identifies co-expressed differential genes significantly associated with the disease and analyzes immune response and inflammatory pathways that may be associated with regulation. Secondly, hub genes were identified by PPI network analysis, and the key genes were cross-analyzed using machine learning methods, and their potential utility as biomarkers or therapeutic targets was verified using ROC curves. Thirdly, the inflammatory and immune response signaling pathways of the two key genes were analyzed for immune infiltration correlation. Finally, possible interactions between the drugs and the expressed proteins of the key genes were modeled to provide targets for further experimental validation and clinical evaluation.

The following discussion of parts of the paper may be confusing to the reader

1. What is the relationship between MYD88/S100A12 and Kawasaki disease in previous studies? Is there any literature review? In other words, how do the authors looking at the simulation findings if they are similar to the findings of previous studies?

2. Results 7, ‘MYD88 showed significant positive correlations with M1 macrophages, activated NK cells, and regulatory T cells (Tregs)’, does not match Figure 7C, please check.

3. Results 7, ‘S100A12 exhibited significant positive correlations with M1 macrophages, regulatory T cells (Tregs), activated NK cells, and CD8 T cells, and negative correlations with CD4 naïve T cells’, does not match Figure 7D, please check.

4. Figure 7A, ‘Immune Cell’, ‘activated’ is incomplete, please check.

5. Figure 7B, ‘cKD’, ‘healthy’ is incomplete, please check.

Reviewer #2: Major:

the focus is on the molecular mechanisms of pyroptosis in KD patients. The author conducted an impressive bioinformatics analysis and identified two top candidate genes, S100A12 and MYD88, proposing them as potential diagnostic biomarkers. They also simulated potential drugs targeting these genes. However, the validation part, in my view, is not sufficient.

The data used is transcriptomic, so I would prefer to see protein-level expression of S100A12 and MYD88, either from wet lab confirmation, such as Western blotting or other database.

From clinical study perceptive, The sample information can be briefly described especially the clinical characteristics. These factors may contribute the expression profile difference. And more sample and diversity data are also encouraged.

From a biological perspective, I believe it would be valuable to include expression profiles from KD patients who received IVIG treatment (a common therapy) as part of the validation. It would be interesting to observe whether the protein levels of S100A12 and MYD88 change in response to treatment.

Minor:

Figure 1G could further clarify the different portions—specifically, which are DEGs, which are genes retrieved from databases, and which are identified through your analysis. The label "pyroptosis" is too close to the WGCNA module; it would be better if the font style matched that of Figure 4G.

Figure 3C: For the Y-axis title, would “expression level” be more appropriate?

6. PLOS authors have the option to publish the peer review history of their article (what does this mean? ). If published, this will include your full peer review and any attached files.

**Do you want your identity to be public for this peer review?** For information about this choice, including consent withdrawal, please see our Privacy Policy .

Reviewer #1: No

Reviewer #2: No

---

## [Author Response · Author response to Decision Letter 1]

8 Apr 2025

Academic Editor's Comment 1:

The authors failed to introduce that COVID and variants of concerns can cause KD. The authors should introduce this.

Author Response:

Thank you for this important suggestion. In response, we have updated the Introduction section to include a brief discussion regarding the association between COVID-19 (and its variants) and Kawasaki disease (KD). Relevant literature (Liu et al., 2021; Liu et al., 2025) has also been cited accordingly.

Academic Editor's Comment 2:

The authors should introduce the function of Myd88. Pattern recognition receptors, such as Myd88, play important roles in recognizing viral nucleic acids as a major class of viral pathogen-associated molecular pattern and triggering intra cellular signaling pathways that culminate in activation of interferon regulatory factor 3 (IRF3) and NF-κB. More references should be cited, with this one (PMID: 38556084) as an example (citing is optional)

Author Response:

We appreciate this valuable comment. In response, we have revised the Discussion section to include a brief introduction to the biological function of MYD88, particularly its role in TLR and IL-1R signaling pathways, which activate downstream IRF3 and NF-κB transcription factors. We have also cited the recommended reference (PMID: 38556084) to support this addition. The revision is located in the Discussion section.

Reviewer #1 Comment 1:

What is the relationship between MYD88/S100A12 and Kawasaki disease in previous studies? Is there any literature review? In other words, how do the authors looking at the simulation findings if they are similar to the findings of previous studies?

Author Response:

Thank you for this valuable suggestion. We have revised the Discussion section to include a brief literature review on the previously reported roles of MYD88 and S100A12 in Kawasaki disease. Specifically, we cited studies demonstrating their upregulation in KD and their involvement in inflammatory and immune responses. We also discussed the consistency between our findings and prior research, which supports the reliability and biological relevance of our analysis.

Reviewer #1 Comment 2:

Results 7, ‘MYD88 showed significant positive correlations with M1 macrophages, activated NK cells, and regulatory T cells (Tregs)’, does not match Figure 7C, please check.

Author Response:

Thank you for pointing out this discrepancy. We have carefully reviewed Figure 7C and the corresponding Results section. The sentence has been corrected to ensure consistency with the figure. The revised statement now accurately reflects the data presented in Figure 7C.

Reviewer #1 Comment 3:

Results 7, ‘S100A12 exhibited significant positive correlations with M1 macrophages, regulatory T cells (Tregs), activated NK cells, and CD8 T cells, and negative correlations with CD4 naïve T cells’, does not match Figure 7D, please check.

Author Response:

Thank you for noting this discrepancy. We have carefully reviewed Figure 7D and the corresponding text. The statement in the Results section has been corrected to accurately reflect the correlation patterns shown in Figure 7D. We apologize for the oversight and have updated the manuscript accordingly.

Reviewer #1 Comment 4:

Figure 7A, ‘Immune Cell’, ‘activated’ is incomplete, please check.

Author Response:

Thank you for pointing out this issue. We have reviewed Figure 7A and found that the label for “Immune Cell”and "activated" was incomplete. We have corrected the label in the revised figure to ensure it is fully displayed and properly aligned with the rest of the content.

Reviewer #1 Comment 5:

Figure 7B, ‘cKD’, ‘healthy’ is incomplete, please check.

Author Response:

Thank you for pointing out this issue. We have reviewed Figure 7B and found that the label for ‘cKD’ and ‘healthy’ was incomplete. We have corrected the label in the revised figure to ensure it is fully displayed and properly aligned with the rest of the content.

Reviewer #2 Comment 1:

The data used is transcriptomic, so I would prefer to see protein-level expression of S100A12 and MYD88, either from wet lab confirmation, such as Western blotting or other database.

Author Response:

We greatly appreciate this valuable suggestion. We acknowledge that confirming expression changes at the protein level would enhance the validity of our findings. However, due to the retrospective nature of our study and current limitations in obtaining suitable biological samples, we were unable to perform wet-lab protein validation such as Western blotting. We agree this is an important next step, and we have clearly stated this limitation and suggested further protein-level validations in future studies in our manuscript.

Reviewer #2 Comment 2:

From clinical study perceptive, The sample information can be briefly described especially the clinical characteristics. These factors may contribute the expression profile difference. And more sample and diversity data are also encouraged.

Author Response:

Thank you for highlighting this important point. We fully agree that detailed clinical information and increased sample diversity would enhance the interpretation of expression differences. Unfortunately, due to the publicly available datasets used in this analysis, we are limited to the clinical data provided by the original authors, which is relatively brief. We have clearly acknowledged this limitation in the revised manuscript and recommended that future studies include detailed clinical information and a broader range of samples to validate our findings further.

Reviewer #2 Comment 3:

From a biological perspective, I believe it would be valuable to include expression profiles from KD patients who received IVIG treatment (a common therapy) as part of the validation. It would be interesting to observe whether the protein levels of S100A12 and MYD88 change in response to treatment.

Author Response:

Thank you very much for this constructive suggestion. In response, we conducted an additional validation using transcriptomic data (dataset GSE16797) from IVIG-responsive KD patients. The results demonstrated significant downregulation of both MYD88 and S100A12 expression after IVIG treatment (Figure 5E and 5F), supporting their involvement in KD inflammation and their potential as markers for treatment response. Although protein-level validation remains an important future direction, our current transcript-level validation supports the biological relevance of these genes in the treatment response context.

Reviewer #2 Comment 4:

Figure 1G could further clarify the different portions—specifically, which are DEGs, which are genes retrieved from databases, and which are identified through your analysis. The label "pyroptosis" is too close to the WGCNA module; it would be better if the font style matched that of Figure 4G.

Author Response:

Thank you for the helpful suggestion. We have updated Figure 1G to clearly label the different portions of the Venn diagram. The section representing differentially expressed genes (DEGs) is now labeled accordingly, as well as the sections for genes retrieved from external databases and novel genes identified through our analysis. These clarifications should improve the figure’s clarity and interpretation.

Reviewer #2 Comment 5:

Figure 3C: For the Y-axis title, would “expression level” be more appropriate?

Author Response:

Thank you for your suggestion. We have updated the Y-axis title in Figure 3C to “Expression level” to improve clarity and better reflect the data being presented. We believe this change enhances the figure's readability.

---

## [Editor Report · Decision Letter 1]

11 Apr 2025

Mechanistic Role of Pyroptosis in Kawasaki disease: An Integrative Bioinformatics Analysis of Immune Dysregulation, Machine Learning-Based Biomarker Discovery, WGCNA, and Drug Repurposing Insights

PONE-D-25-12866R1

Dear Dr. Wang,

We’re pleased to inform you that your manuscript has been judged scientifically suitable for publication and will be formally accepted for publication once it meets all outstanding technical requirements.

Kind regards,

Benjamin M. Liu, MBBS, PhD, D(ABMM), MB(ASCP)

Academic Editor

PLOS ONE
---

## [Editor Report · Acceptance letter]

PONE-D-25-12866R1

PLOS ONE

Dear Dr. wang,

I'm pleased to inform you that your manuscript has been deemed suitable for publication in PLOS ONE. Congratulations! Your manuscript is now being handed over to our production team.

Kind regards,

on behalf of

Dr. Benjamin M. Liu

Academic Editor

PLOS ONE